# The Evolutionary Dynamics of the Mitochondrial tRNA in the Cichlid Fish Family

**DOI:** 10.3390/biology11101522

**Published:** 2022-10-18

**Authors:** Yosur G. Fiteha, Mahmoud Magdy

**Affiliations:** Genetics Department, Faculty of Agriculture, Ain Shams University, Cairo 11241, Egypt

**Keywords:** cichlid, haplotilapiine, mitochondrial genome, tRNAs, secondary structure

## Abstract

**Simple Summary:**

Cichlids are a unique example of fish diversity and species richness which have been explained by sympatric speciation at different freshwater sources in Africa. The mitochondria contribute to cell vitality by providing energy. It contains a circular genome with an established translation system that is spatially independent of the cytosolic counterpart. The current study aimed to investigate the evolutionary dynamics of the mitochondrial tRNA and its role in cichlids’ diversity. The available cichlid mitogenomes in the public database were filtered, in addition to newly sequenced accessions from a specific cichlid group known as the haplotilapiine lineage that is widely distributed in the Egyptian sector of the Nile River. Based on the comparative analysis of mitogenomic data, we identified 22 tRNA genes, in which a single gene was D-armless, while the cloverleaf secondary structure subdivided into stem-loop formations was predicted and used to define the levels of genetic divergence for the remained tRNAs. Peculiarly, in cichlids, the formation known as “T-arm” showed the lowest polymorphism levels among other structures in contrast to other organisms (e.g., scorpions). Comparing the whole family to the specific haplotilapiine lineage showed that the tryptophan tRNA was the most conserved tRNA, with signatures of possible purifying selection.

**Abstract:**

The mitochondrial transfer RNA genes (tRNAs) attract more attention due to their highly dynamic and rapidly evolving nature. The current study aimed to detect and evaluate the dynamics, characteristic patterns, and variations of mitochondrial tRNAs. The study was conducted in two main parts: first, the published mitogenomic sequences of cichlids mt tRNAs have been filtered. Second, the filtered mitochondrial tRNA and additional new mitogenomes representing the most prevalent Egyptian tilapiine were compared and analyzed. Our results revealed that all 22 tRNAs of cichlids folded into a classical cloverleaf secondary structure with four domains, except for trnS^GCU^, missing the D domain in all cichlids. When consensus tRNAs were compared, most of the mutations were observed in the trnP at nucleotide levels (substitutions and indels), in contrast to trnL^UAA^. From a structural perspective, the anticodon loop and T-loop formations were the most conserved structures among all parts of the tRNA in contrast to the A-stem and D-loop formations. The trnW was the lowest polymorphic unneutral tRNA among all cichlids (both the family and the haplotilapiine lineage), in contrast with the neutral trnD that was extremely polymorphic among and within the haplotilapiine lineage species compared to other cichlids species. From a phylogenetic perspective, the trnC was extremely hypervariable and neutral tRNA in both haplotilapiine lineage and cichlids but was unable to report correct phylogenetic signal for the cichlids. In contrast to trnI and trnY, less variable neutral tRNAs that were able to cluster the haplotilapiine lineage and cichlids species as previously reported. By observing the DNA polymorphism in the coding DNA sequences (CDS), the highest affected amino acid by non-synonymous mutations was isoleucine and was equally mutated to valine and vice versa; no correlation between mutations in CDS and tRNAs was statistically found. The current study provides an insight into the mitochondrial tRNA evolution and its effect on the cichlid diversity and speciation model at the maternal level.

## 1. Introduction

Cichlid fishes have become well-known model species in studying evolutionary biology, because of their exceptionally diverse morphology, behavior, and ecology [1,2]. Tilapia is a common name for several species that belong to three genera, namely, *Oreochromis*, *Sarotherodon*, and *Coptodon* (syn. *Tilapia*) [3]. They are highly abundant in natural sources and are considered the second most edible fishes globally [4]. Furthermore, tilapias have become the most important fish species in freshwater and brackish water aquaculture in some countries due to their excellent growth rate, resistance to diseases, environmental adaptability, ease of breeding, and market acceptability [5]. Despite the familiarity and great economic and ecological significance, few molecular phylogenetic studies focus on the haplotilapiine lineage (e.g., [6]).

In the last decade, the mitochondrial genome (mtDNA) of Metazoa has been regarded as the marker of choice for species identification, molecular phylogenetic, and population studies. It has been widely used for the resolution of taxonomic controversies. In animals, the mitochondrial genome is generally a small, circular molecule of 15–20 kb that usually encodes 37 genes, including 13 protein-coding genes, two ribosomal RNAs (rRNAs), and 22 transfer RNAs (tRNA) [7]. The small size of the molecule, the presence of genes/regions evolving at different rates, uniparental inheritance, and the absence of recombination make this molecule an effective and easy-to-use phylogenetic marker [8,9].

Transfer RNAs (tRNAs) play an essential role in protein biosynthesis and post-transcriptional regulation in all living organisms. In the gene translation process, a molecule of tRNA should be bound to the convenient amino acid, which primarily depends on the tRNA structure [10]. The classically accepted secondary structure for canonical mt-tRNAs is a “cloverleaf” consisting of four stems and three loops [11,12,13]. The anticodon loop contains the three base anticodons specific to the appropriate mRNA codon, ensuring that the correct amino acid is added during translation. The acceptor stem undergoes maturation and is charged with the appropriate amino acid [11,14]. Transfer RNA genes attract more attention because the tRNA gene family evolution in the mt genome displays intricate patterns characterized by huge variability, including multiplication, gene rearrangement, loss/deletion, and duplication compared to nuclear tRNA [15,16]. Additionally, the mitochondrial tRNAs deviate from the classical cloverleaf tRNA form (e.g., nematode) and accumulate polymorphism associated with pathogenesis and syndromes compared to the nuclear tRNAs in humans [17].

The objective of the current study is to: (a) detect and validate tRNA genetic variation in the Cichlidae family and the haplotilapiine subcluster; (b) study the impact of these variations on the secondary structure form; and (c) validate and evaluate the variable tRNAs from genetic divergence and evolutionary perspectives.

## 2. Materials and Methods

### 2.1. Sample Collection and DNA Extraction

Sampling for the current study was performed from different waterbodies in the Nile River network in Egypt during summer 2020. Three different haplotilapiine species were collected from fishermen in triplicate; *Oreochromis niloticus* was collected from Qarun lake (29.4840° N, 30.6545° E), *Coptodon zillii* from Nasser Lake (22.7395° N, 32.1973° E), and *Sarotherodon galilaeus* from Manzala Lake (31.3306° N, 32.0497° E). Specimen identification was conducted by checking morphological characteristics following Trewavas [18]. Tissue samples from these individuals were preserved in 100% ethanol and stored at −20 °C before use. Total genomic DNA was extracted from tissue samples following the manual protocol of Li et al. [19]. Sample integrity was checked by agarose gel (1%) electrophoresis and visualized under UV-light using the Ingenius3 Gel documentation system (Syngene, Cambridge, UK).

### 2.2. Bioinformatics Analysis

Cichlids mt genome sequences have been imported from available fully sequenced genomes in the GenBank database. All cichlid mitogenomes were subjected to reannotation that was carried out using GeSeq-Annotation of Organellar Genomes [20]. Additionally, any potential recombination signals in cichlid mitogenomes were assessed using the Recombination Detection Program software (RDP4) [21]. Finally, a heatmap was generated based on the complete mt genomes, tRNAs, CDS, and intergenic spacer (IGS) separately using Heatmapper (http://www.heatmapper.ca/) to represent and cluster the analyzed cichlid species based on the genetic dissimilarity relationship.

The tRNAScan-SE V2.0 software [22] was employed to determine whether all the downloaded sequences encoded tRNA or not, and the resulting novel tRNA structures were reannotated in the original files. The parameters of tRNAScan-SE V2.0 were set as vertebrate mitochondrial for “sequence source”, the default for “search mode”, formatted (FASTA) for “query sequences”, and universal for “genetic code” for tRNA isotype prediction. Furthermore, VARNA V3.93 software [23] was used as companion software to visualize the secondary structure of tRNAs.

A complete set of tRNA genes from the mt genomes of cichlids were extracted, each gene was aligned individually using the MAFFT algorithm [24]. Then, the consensus sequence of each tRNA alignment was generated following the majority role. Consensus tRNA sequences were used as a reference for single nucleotide polymorphism (SNPs or substitutions) calling among cichlid species, and aligned manually among all tRNAs, preserving the tRNA structural domains (i.e., acceptor “A-stem”, “D” -loop and -stem, anticodon “An” -loop and -stem, variable region “V-loop”, “T” -loop and -stem, and central connecting “CC” -loop) to avoid incorrect gap insertions. Additionally, the neutrality test was estimated using Tajima’s D test [25] for all CDS and tRNAs using DnaSP V6 [26]. The tRNAs alignments were analyzed with MEGA X software [27] to find the transition/transversion bias. The statistical parameters used to analyze the transition/transversion bias (*R*) were set at the maximum composite likelihood method, based on the Tamura–Nei model, while partial deletion of gaps/missing data treatment was set with site coverage cutoff equal to 95%. The *R* value was estimated for each tRNA (*R*_tRNA_) and for the total CDS (*R*_CDS_) to be used as a cutoff to the *R*_tRNA_ values.

A “substitution-quantification” approach was followed to define and quantify the conserved and polymorphic sites for each tRNA gene. This approach treated each position as a separate locus. To this end, indels and nucleotide polymorphism were measured among the tRNA genes and defined per each structural domain. First, the frequent dominant base was investigated at each position of the tRNA alignment. The most frequent base in a single position was considered the dominant base (a.k.a., conserved base “Cn”). In contrast, other bases were labeled “Sn” as transition base when the base was identified of the same group of nitrogenous bases (i.e., purine or pyrimidine), and “Vn” for the transversion when the base was identified from the different nitrogenous bases group. If a site contained an indel, the position was labeled “Dn” for deletion when gaps were less frequent than any base substitution; otherwise, the site was considered as an insertion “In”, and the inserted bases were categorized using the previous role.

When two or more bases recorded an evenly equal number of occurrences at one position, the dominant base was defined using the standard IUPAC degeneracy codes. In this case, the labeled categories were defined according to the dominant degenerated base. For example, in variant sequences at one position, A and G recorded an equal number of SNPs, the dominant base was defined as R. In this case, variant sequences with A or G were labeled “Cn” while the other was labeled “Sn”, while any other variant sequence was labeled either Vn, In, Dn, or in combination, following the same previous rule. Then the total count of each labeled category (either Cn, Sn, Vn, In, Dn, or in combination) was summed and sorted decreasingly: (a) for each species per tRNA; and (b) for each tRNA per structural domain. The abbreviated letters were not illustrated in the results to avoid confusion with the amino acids letter codes.

Second, each position per sequence was weighted by a fixed categorial number (0.00, 0.25, 0.50, 0.75, and 1.00) based on each labeled category total count. Accordingly, the highest weight was given to the labeled category with the lowest total count and vice versa; however, if the type of mutations was either transitions, transversions, or both, only, a fixed weight was used as 0.25 for every transition and 0.50 for every transversion. The estimated averages based on the weighted values were: (a) used to plot an alluvial diagram using the RAWGraphs online tool [28] to demonstrate the relationship between the distribution of mutations among tRNA structural domains; (b) used to estimate the parameters required for the box plot graph using an Excel template (http://www.vertex42.com) among different tRNA based on the cichlid species, and (c) normalized using the z-score transformation method to compare the average weighted total variation recorded for the cichlid family versus the newly sequenced tilapiine species.

Changes in the amino acid occurring along the coding DNA sequence were examined using Geneious R10 (Biomatters, Auckland, New Zealand) [29]. The abbreviations and single-letter codes were used for the 20 amino acids. The model of nucleotide substitution was used at the codon level to obtain the nucleotide-level information in CDs and knowledge of the genetic code and hence the amino acid-level information of synonymous and nonsynonymous nucleotide substitutions. Then, the SNPs among the CDS were classified and analyzed according to their functional and evolutionary categories.

### 2.3. Primer Design, PCR, and Wet-Lab Validation

All specimens were subject to validation by PCR using universal primers that targeted the COI gene (COI-FF2D1: 5′-TTCTCCACCAACCACAARGAYATYGG-3′ and COI-FR1D-1: 5′-CACCTCAGGGTGTCCGAARAAYCARAA-3′), which was amplified as a reference gene to overcome the problem of species misidentification. In addition, selected tRNA genes were amplified by new primers designed using the primer design tool in Geneious R10, according to the alignment of mtDNA of *O. niloticus* (NC_013663) and *C. zillii* (NC_026110) from the National Center for Biotechnical Information (NCBI). PCR was carried out using EasyTaq DNA Polymerase (TransGen Biotech, Beijing, China). Amplifications were performed in a total volume of 25 μL, containing 2.5 μL of EasyTaq 10X buffer, 2 μL of dNTPs (10 μM), 1 μL of each primer (forward and reverse, each of 10 μM), 0.2 μL of EasyTaq DNA polymerase, and 1 μL of extracted DNA (~100 ng/μL). The thermocycling program was performed under the following conditions: an initial step of denaturation at 95 °C/5 min, followed by 32 cycles of 94 °C/1 min, annealing temperature at 50–55 °C/30 s (depending on the primer’s Tm), extension at 72 °C/90 s, and a final extension segment at 72 °C/10 min. Subsequently, the amplifications of PCR were visualized on 1.5% agarose gels stained with ethidium bromide, and successful amplifications were purified by spin column using EasyPure PCR Purification Kit (TransGen Biotech, Beijing, China) following the manufacturer’s instructions. The purified samples were commercially sequenced bidirectionally (Macrogen Inc., Seoul, South Korea). Sequences of both directions were evaluated, trimmed for quality, assembled, and identified using the BLAST search tool in the NCBI database applying default parameters. Alignments of the target sequences and the BLAST query results were performed using the MAFFT aligner implemented in Geneious R10.

## 3. Results

### 3.1. Characterization of Cichlids Mitochondrial Genomes

The genomic survey was performed based on available sequences of known complete mitochondrial genomes deposited in public databases. The list of cichlids species used for this study is given in Appendix A. We have analyzed a dataset comprising 95 cichlids’ mitogenomes, 17 were removed from the compilation due to significant variations regarding the other sequences, or annotation errors either identifying tRNA genes, being identified as a recombinant sequence, or both.

The structure of all retained cichlid mitogenomes was a typical circular molecule, with an average length 16,583 ± ~66 bp. All species contained 13 protein-coding genes (ATP6, ATP8, COI–III, Cytb, ND1–6, and ND4L), 22 tRNA genes, two ribosomal RNA genes (12S rRNA and 16S rRNA), and a putative control region (D-loop). The mitogenomes’ details are shown in Appendix A.

A total of 78 cichlids’ mitogenomes were aligned and utilized to construct a maximum-likelihood phylogenetic tree based on complete mt genomes, tRNAs, CDS, and IGS data sets separately to clarify the interspecific relationships between cichlids. A member of the Mugilidae family (*Liza haematocheilus*, NC_024531) was added to serve as an outgroup. The genetic distance estimated from the phylogenetic analysis was visualized for each set as a heatmap (Figure 1). Among cichlids, a large block of highly homogenized species was defined, this block included haplotilapiine species (i.e., genera *Oreochromis*, *Sarotherodon*, *Coptodon*, and *Stomatepia*) with some outliers from other groups (indicated by a red arrow in Figure 1). Within the block, the *Oreochromis* species were separated into two smaller blocks, one included only members of the *Oreochromis* genus, while the other showed a mixed clustering among species from both *Oreochromis* and *Sarotherodon* genera. This subdivision was found for all data sets, except for the IGS data (Figure 1). Within the haplotilapiine, the relationships among inner clusters were highly supported (i.e., >50% bootstrap values for many branches; Appendix A).

### 3.2. Diversification of tRNA Secondary Structure

The tRNA sequences elucidated from the retained cichlid mitogenomes were analyzed for their conserved secondary structural form “cloverleaf”. The secondary structure of 22 mt-tRNAs in all cichlid mitogenomes was estimated and compared. Eight tRNA genes (trnQ, trnA, trnN, trnC, trnY, trnS^UGA^, trnE, and trnP) were encoded by the reverse strand, whereas the remaining tRNAs were encoded on the forward strand. All investigated cichlids species were shown to possess at least one anticodon type for each kind of tRNA. The sequences of all tRNAs were folded into the canonical cloverleaf secondary structure composed of four domains and a short variable loop, except for trnS^GCU^, which has the most deviating structure of them all, with a missing D-arm. The tRNAs were scattered around the mitogenome and ranged from 67 (trnC and trnS^GCU^) to 74 bp (trnL^UAA^) in size (Appendix A). The results of structure analysis showed that the length of stems in most of the tRNAs was fixed (A-stem = 7 bp, An-stem = 5 bp, and T-stem = 5 bp), except for the D-stem that varied from 3 to 4 bp (Figure 2). Regarding the loops, whereas the length of the An-loop was fixed (seven nucleotides) in most of the tRNAs, the other loops were variable in length. The T-loop mostly recorded seven nucleotides with a range of 7–9 nt while the V-loop ranged from 4 to 6 nt in length and D-loop was extremely variable with a range of 3–10 nt (Figure 2).

### 3.3. Mitochondrial tRNAs Polymorphism among Cichlid Species

#### 3.3.1. Quantified Polymorphism among mt-tRNAs

Based on the alignments between all tRNAs consensus sequence obtained by the “bases match at least 95% of the sequence” rule for each tRNA among all cichlids, several mutations along the secondary structure of the tRNAs were counted by mutation type and location. Among consensus tRNA alignment, the average percentage of recorded conserved sites was 51 ± 5% versus 40 ± 5% for SNPs and 9 ± 3% for indels. The average mutation rates were recorded as ~40 bp for conserved sites, ~16 bp for transversions, ~14 for transition sites, ~5 bp for deletions, and ~3 bp for insertions, and weighted as 0.00, 0.25, 0.50, 0.75, and 1.00, respectively. After applying for the weight role, the highest total number of weighted SNPs was found in trnP (9.36) followed by the trnI (8.07), in contrast to the trnL^UAA^ (2.93) and trnE (2.98). As for indels, the highest total number of weighted indels was observed in the trnY (8.07) followed by trnP (7.32), in contrast to the trnK (2.97).

From tRNA structure perspective, the highest total number of weighted SNPs was found in the A-stem (36.91, average among tRNAs = 1.75 ± 1.0), in contrast to the CC loop (3, average among tRNAs = 0.14 ± 0.16) and T-loop (5.83, average among tRNAs = 0.27 ± 0.22). As for indels, the highest total number of indels polymorphism was observed in the D-loop (43.64, average among tRNAs 2.07 ± 0.67) followed by V-loop (37.42, average among tRNAs 1.78 ± 0.87), in contrast to the T-loop (2.75, average among tRNAs = 0.13 ± 0.15), the An-loop (3, average among tRNAs = 0.14 ± 0.16) and T-stem (4, average among tRNAs = 0.19 ± 0.23).

#### 3.3.2. Quantified Polymorphism of Each mt-tRNA among Cichlids

The boxplot was utilized to identify the accumulated polymorphism based on alignments of each tRNA among cichlids (Figure 3B). The overall average ratio of weighted mutations of each tRNA was 0.012 ± 0.006. The highest weighted number of mutations was observed in trnP (0.025 ± 0.008), trnY (0.024 ± 0.012), trnC (0.023 ± 0.009), and trnH (0.20 ± 0.007), while the contrary was observed in trnF (0.0050 ± 0.006), trnE (0.0055 ± 0.008), and trnW (0.0058 ± 0.008), respectively.

#### 3.3.3. The Transition/Transversion Bias and Phylogenetics

The transition/transversion substitution bias was estimated based on the observed nucleotide counts. The observed nucleotide ratios among all tRNAs were 25.44% (A), 28.30% (T/U), 22.50% (C), and 23.76% (G) with CG:AT = 53.74:46.26%. The transition/transversion rate ratios were k_purines_ = 2.367 and k_pyrimidines_ = 2.136. The overall transition/transversion bias was *R* = 1.117. The total *R*_CDS_ value was 4.92, which was used as a cutoff to the *R*_tRNA_ values, which recorded an average of 7.30 ± 7.23. The highest bias was 33.72 recorded for the trnS, in contrast to trnG of 1.50 *R* value. The retained tRNAs after applying the cutoff were categorized in two groups, one recorded double the *R*_CDS_ value (trnS, trnL^UAG^, trnR and trnW) while the other recorded less than double (trnA, trnE, trnH, and trnK; Appendix A; Appendix A).

Using the maximum likelihood method, an evolutionary tree was constructed among the aligned tRNAs. Due to the lack of a proper outgroup, the tree was rooted using the longest untransformed branch supported by the highest substitution rate and bootstrap values. The three clades (I, II, and III; Appendix A) contained eight, five, and eight tRNAs. The most distant tRNAs were trnV and trnW, in subclade 1 and trnG and trnL^UAG^ in subclade 2 of clade I; and were trnR and trnQ for clades II and III, respectively. When the transition/transversion bias was considered, trnW, in subclade 1 and trnL^UAG^ in subclade 2 of clade I; and was trnR for clade I and trnS and trnN for clade III (Appendix A).

### 3.4. Cichlids vs. Haplotilapiine

The total numbers of mutations in each tRNA gene were examined in all cichlids species (Ci) versus a range of haplotilapiine lineage (Ht; Figure 4). The normalized variation patterns by the z-score method were divided into four categories according to how many variants accumulated in each tRNA gene (levels of hyper- or hypo-variability). The 1st category of hyper-variable tRNAs in both cichlids and haplotilapiine lineage were trnC, trnY, trnP, trnG, and trnI. On the contrary, the 2nd category of hypo-variable tRNAs in both cichlids and haplotilapiine lineage were trnA, trnE, trnF, trnK, trnL^UAG^, trnN, and trnQ. The 3rd category of the tRNA genes identified as hyper-variable in cichlids and hypo-variable in haplotilapiine lineage were trnH and trnM. Finally, th^e^ 4th category for tRNA genes identified as hypo-variable in cichlids but hypervariable in haplotilapiine lineage were trnD, trnL^TAA^, trnR, trnS, trnT, trnV, and trnW. The Tajima D values ranged from 0.0383 to −2.069 and were found to be significantly deviated from zero for trnE, trnG, and trnW in Ci and for trnH, trnV, trnT, and trnW for Ht (Appendix A).

### 3.5. The tRNA in Egyptian Haplotilapiine

Based on the previous categories, the trnC and trnD were selected to be sequenced to perform further comparative secondary structure analysis across different haplotilapiine specimens (i.e., samples from *O. niloticus*, *C. zillii*, and *S. galilaeus* species) that represent the widely distributed tilapia species found throughout Nile River and connected waterbodies in Egypt. The trnC was extremely hypervariable in both haplotilapiine lineage and all cichlids (1st category) with a remarkably smaller D-loop structure compared to all other tRNAs, while the trnD was the only tRNA gene that displayed extreme hypervariability in haplotilapiine lineage in contrast to all cichlids (4th category). After that, the secondary structure of the selected tRNAs was examined, validated, and cross-checked with the general tRNA pattern in haplotilapiine lineage and cichlids (i.e., presented by the consensus sequences of each group).

The trnC varied in total length from 67 bp in cichlids to 66 bp in haplotilapiine (Figure 5A), indicating that trnC was the extreme example of shortening. Additionally, a degenerated nucleotide (H) in position No. 15 was detected in some cichlid species. In the trnC sequence, 59 sites were identical (89.4% of conservation) for the three haplotilapiine species. Mutations were scattered throughout the gene while the D-loop was composed of 3 bp only. Seven mutations were identified, four of which were contributed by *C. zillii* (Figure 5A). All mutations discovered between the three species were transition, which was found in the following positions: 16, 19, 20, 25, 27, 47, and 62. Two base-pairing mismatches were identified at positions T_5_ • G_61_ and G_10_ • T_20_. Despite the detected mutations, both trnD and trnC maximum likelihood phylogenetic trees were rooted by *Liza haematocheila* and were unable to differentiate between the haplotilapiine species.

The trnD was identical in length (73 bp) compared to all cichlid species (Figure 5B), and the molecular analysis revealed that 64 positions were identical (87.7% of conservation) for the three haplotilapiine species. The mutations were mainly localized in the D-loop and the T-loop. The three specimens differed at nine two-variant sites: G → A transition at positions 15, 57, and 64; T → C transition at position 20; C → T transition at position 21; A → T transversions at position 56, and A → G transition at positions 59 and 73. Only one three-variant site at position 17 represented the maximum interspecific variation among the three haplotilapiine species. A single mispairing mutation was detected in *C. zillii* at position C_50_ • A_64_ (Figure 5B).

### 3.6. Amino Acid Change

One of the methods to measure the effects of natural selection in molecular evolution is to estimate the rates of synonymous and nonsynonymous substitutions. The mitogenome of *Oreochromis* sp. (NC_009057.1) was set as the reference genome sequence to call SNPs. A total of 2091 amino acid changes constitute different mutation types: 1837 synonymous (no effect), 237 nonsynonymous (amino acid substitution), one truncation (COX1), four deletions (COX1), and two frameshifts (COX1 and ND5) were retrieved from the CDS region. However, Tajima’s D values were not different from zero for all CDS regions (Appendix A).

Based on the nonsynonymous mutations, two types of changes were recorded for each amino acid. A forward mutation causes a change toward the translation of specific amino acid, and a backward mutation causes a change reverted from the translation of that specific amino acid. The highest forward mutations were observed for isoleucine “I” by 41 changes in Ci and 15 changes in Ht, followed by A, T, V, S, and L in both cichlids and haplotilapiine lineages. However, the changes towards I came from L, V, M, T, and S in cichlids and from V and T in haplotilapiine. In cichlids, the amino acids E and W recorded no changes versus C, D, E, K, and W in haplotilapiine lineage. The lowest forward mutations were observed for C, D, and Q (two changes) followed by R (two changes) in cichlids and Q, R, and Y (one change) for haplotilapiine (see the complete list in Appendix A; Figure 6). Reversely, the highest backward mutations in cichlids were observed for T by 33 changes, followed by S and L, in contrast to C, and R. In the haplotilapiine lineage, the highest backward mutations were observed for V, followed by T; in contrast to C, G, K, W, and Y equally, with zero change. The amino acids D, Q, W, T, and Y showed an approximate equilibrium between forward and backward mutations in cichlids and amino acids A, C, F, M, K, P, Q, and W in haplotilapiine (Figure 6).

After estimating the R-value based on simple linear correlation analysis and matrix-based correlation analysis (partial Mantel test), we found no significant results (*p*-value > 0.05) between any type of the accumulated mutations and the changes in amino acid frequency due to forward or reverse mutations.

## 4. Discussion

All the cichlids mitogenomes deposited in the GenBank database were used in the current study, many were published, while others were unpublished that be used with caution after some filtration and further processing. Based on the initial inspection of the mitogenomic data, several annotation errors, recombinant sequences, and significant heteroplasmy were detected. In cichlids, hybrids were reported repeatedly (e.g., gray and red tilapia [30]), that would be one reason for the presence of mislabeled accessions as morphologically appearance is different from the maternal species, while the maternally inherited mitochondrial DNA shows only the maternal genetic signature (e.g., *Oreochromis aureus*; GenBank accession: NC_013750). Another explanation might be due to the usage of the next-generation sequencing technique, in which the assembly might be affected by the nuclear mtDNA homologous sequences that cause the false heteroplasmy [31]. Therefore, the combination of phylogenetic analysis with BLAST-based identification would help to improve the species identification and thus, avoid such incongruences in species clustering. The current preliminary analysis provides a filtration procedure that can be followed for a similar type of investigation.

By applying the tRNA secondary structure prediction, four-armed cloverleaf secondary, and L-shaped tertiary structures were clarified and subdivided into stem-loops formations. Stem-loops are believed to occur during single-stranding events when inverted repeats meet to form a region of pairing (the stem) surmounted by their interceding sequence (the loop [32]). The current study found that the D-stem was the only stem varied in size among cichlids species. In contrast to loops, where the only fixed loop in length was the anticodon loop versus the D-loop. Loop regions of stem-loop secondary structures are often associated with hot spots for mutation, affecting both nucleotide substitutions and indel events that consequently affect its length (e.g., [33]). In cichlids, most substitutions were observed in stems (i.e., A-stem), while indels were more frequent in loops (i.e., D- and V-loops). Although indels are most common in the terminal loops, they may occur anywhere along the secondary structure [32]. In some cases, small segments of the stem itself would be deleted, decreasing the stem length, though perhaps not to an extent that would annihilate possible secondary structure formation [32]. In contrast to plants and fungi, almost all trnS for AGY/N codons in metazoan mitogenomes lack the D-arm, the D-armless tRNA is believed to first emerged after the metazoan branching [34]. In cichlids’ mitogenomes, the trnS^GCU^ has a missing D-arm, marking the most deviating structure among all tRNAs [34]. Even though the trnC showed the same length as trnS^GUC^ but form a complete cloverleaf structure without any missing arms. Several mt-genomes from the acariform lineage and scorpions, possess many tRNA genes that lack either D-arm or T-arm sequence [35]. On the contrary, in cichlids, the T-arm (stem and loop) in addition to the anticodon stem showed the lowest polymorphic structure of all. In this work, secondary structure prediction was used to improve the alignment to combine both secondary structure information and maximum likelihood in a straightforward way that could help and improve the inter-tRNA comparative analysis.

The cichlids are a peculiar example of fish diversity at the phenotypic level, however, at the cytogenetic level, the cichlids recorded a moderate chromosomal divergence rate among Perciformes [36]. Sympatric speciation through sexual selection explained the outstanding species richness in the cichlid family at different locations [37,38,39], where genetic polymorphism plays a key role to initiate speciation followed by either reproductive isolation, sexual selection preference, or both. We compared the haplotilapiine lineage (Ht) to the cichlid family (Ci) to resolve possible evolutionary pressures on the mt-tRNA sequences related to the speciation diversity within the cichlid family at the maternal level. Four categories were defined according to their weighted genetic variability. The presence of hyper-variable (1st category) and hypo-variable (2nd category) regions in both Ht and Ci reflect a homogeneous evolutionary force at different taxonomical levels within the cichlid family. However, the presence of contrasting variability between Ht and Ci required an explanation. The trnP repeatedly appeared among the highly polymorphic tRNAs, in humans, an anti-codon swap and a duplication event of the trnP were associated with myopathy and dilated cardiomyopathy, respectively [40,41]. In contrast to trnE, trnW both recorded the lowest polymorphic level and were found to be unneutral. Under a model of neutral evolution, Tajima’s D was calculated for the CDS regions and tRNAs for cichlids species and haplotilapiine group, which allowed us to identify potential genes under selection. Tajima’s D was not different from zero for all CDS regions, indicating that the observed frequency of polymorphism was lower than expected and the site frequency spectrum did not deviate from that predicted by neutral theory [25]. The majority of tRNA genes have conserved codons and will probably not undergo selection, but some of the Tajima’s D values based on tRNAs distribution were found to deviate significantly from the normal for both cichlids and haplotilapiine (Appendix A). Significantly negative Tajima’s D values were estimated for trnE, trnG, and trnW in Ci, and trnH, trnT, trnV, and trnW in Ht. A negative Tajima’s D means an excess of low-frequency polymorphisms relative to expectation, indicating either population size expansion (e.g., after a bottleneck or a selective sweep), purifying selection, or both [25].

The trnW (tryptophan) was one of the tRNAs with the lowest polymorphism rate and was found to be unneutral in both Ci and Ht. It showed the lowest forward and backward amino acid changes among all other tRNAs. Moreover, the trnW recorded a transition and transversion bias value double that of the R_CDS_ and represent the most diverted within the subcluster 2 of clade I formed by the phylogenetic tree among all tRNAs. The results suggest a strong selection against mutations in trnW structure, as well as mutations affecting the amino acid changes among the cichlid species. On the contrary, the trnH (histidine) recorded a contrasting genetic variability between Ci and Ht, the latter in which it was extremely low and significantly deviated from the neutral state of evolution. The mitochondria are inherited without a recombination process comparable to the meiosis of the nuclear DNA, its genes are either single, complete sets, or both, of coding genes referred to as “supergenes” [42], thus coding genes and tRNAs located at proximate locations are linked. Even without evolutionary importance, a gene could be affected when another region behaves in a non-neutral manner. When a neutral allele is favored by a linked gene under positive selection, the process is called genetic hitchhiking [43]. The trnH is located between ND4 and ND5, both recorded the most variable CDSs among cichlids (Appendix A). The trnH was associated with important evolutionary divergences, in crab (infraorder: Brachyura), trnH is translocated between nad3 and nad5 by tandem duplication/random loss model from its ancestors (order: Decapoda [44]). Moreover, a positive association of trnH to viral infections was previously reported [45]. The presence of specific genetic variation in tRNAs for the haplotilapiine lineage separated from the cichlid family is in accordance with the sympatric speciation model.

After validation by sequencing, the trnD was the only tRNA gene that recorded the highest and extreme genetic variability in haplotilapiine lineage in contrast to all cichlids, and the trnC was extremely hypervariable in both haplotilapiine lineage and cichlids. However, both were neutral, with a lower R value than R_CDS_ among cichlids and none were able to distinguish the haplotilapiine genera. In contrast to trnI and trnY, two other hyper-variable tRNAs in both cichlids and haplotilapiine lineage, both were neutral, with lower R value than R_CDS_ among cichlids, however, both were able to differentiate the haplotilapiine genera with minor differences, where *Oreochromis* genus is closer to *Sarotherodon* than to *Coptodon* (e.g., [6,46]). The reversely proportional relation between transition/transversions bias and level of genetic divergence was previously highlighted [47]. Inferring the phylogenetic affinities and classification within the cichlids has been a longstanding challenge [46]. The reliance on mitochondrial genes in DNA barcoding (e.g., COI [48]) assumes that the mt-based phylogenetic trees reflect the evolutionary dynamics among species. However, being subject to evolutionary pressures (i.e., unneutral) may significantly affect the tree structure [49]. The neutral state of trnC, trnD, trnI, and trnY suggests that DNA substitutions are not affected by any non-random factors [50] and are suitable loci for phylogenetic studies. However, only trnI and trnY showed sufficient variation to reflect the cichlid speciation and specifically the haplotilapiine lineage at the maternal level, thus we propose to concatenate both hypervariable genes to study haplotilapiine and cichlid phylogenetics.

Isoleucine anti-codon is on the edge, through transition mutation in th^e^ 1st codon, trnI will be transformed to trnV, while transition in the last codon will transform it into trnM. Methionine codon “ATG” is the start codon for open reading frames in mitochondrial genes. However, the number of forward mutations is mostly equal to backward mutations between isoleucine and valine (18:19 mutations) and none were recorded for isoleucine to methionine or vice versa. Our finding suggests that a non-functional selection in the mitochondrial genes might be occurring during the speciation of the cichlid family.

## 5. Conclusions

The cichlid family evolution follows the sympatric speciation model, in which the intraspecific genetic polymorphism is the key source for the emergence of new species. Within this context, the evolutionary mechanism of the mitochondrial tRNA was evaluated and inspected among the family members available in the GenBank database, providing useful insights into the cichlids’ diversity at the maternal level. In accordance with metazoan, one D-armless tRNA was detected (namely, trnS^GCU^), while the T-loop secondary structure formation was remarkably conserved. Among cichlids, the manually aligned consensus haplotypes of tRNAs showed that the trnP and trnF were the highest and lowest polymorphic tRNAs, respectively. While within the haplotilapiine lineage, a group of the cichlids inhabiting the northern part of the Nile River, the trnD and trnM were the highest and lowest polymorphic tRNAs, respectively. When both the cichlid family and the haplotilapiine lineage were compared, the trnC and trnF were the highest and lowest polymorphic tRNAs, respectively. When the deviation from neutral evolution was considered, the trnW was the only low-polymorphic unneutral tRNA genes among all, suggesting purifying selection against the tryptophan tRNA variation in cichlids. All the detected polymorphisms in the coding DNA sequences were found neutral and were not statistically correlated to the polymorphism observed in the tRNAs of the studied cichlids. However, most of the non-synonymous mutations were changing the isoleucine to valine and backward. But no changes toward an approximate amino acid methionine (a key amino acid to start open reading frames) were observed, confirming that a non-functional selection in the mitochondrial genes might be occurring during the sympatric speciation of the cichlid family.

## Figures and Tables

**Figure 1 biology-11-01522-f001:**
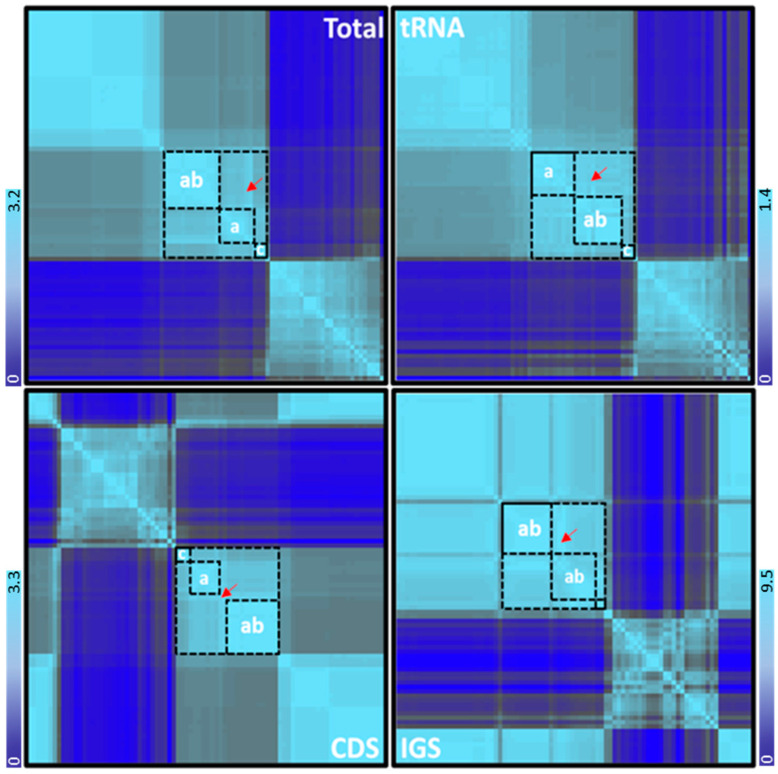
The heatmap represents the correlation between 78 cichlid species. The heatmap is constructed based on the matrix of complete mitochondrial genomes, tRNAs, CDS sequence, and IGS regions, and the phylogenetic analysis was performed by the maximum likelihood method. The haplotilapiine species are homogenized together in the same block, represented in the black box. The main black box is split into three small boxes—a: represents *Oreochromis* genus, ab: represents a mixed group of species (either *Oreochromis*, *Sarotherodon*, or both), and c: represents *Coptodon* genus. The red arrow represents an anomalous branch in the middle of the cluster. An increase of genetic diversity is indicated in light blue and a decrease in dark blue. The intensity of the color corresponds to the strength of the effect, referring to the median of the affected node.

**Figure 2 biology-11-01522-f002:**
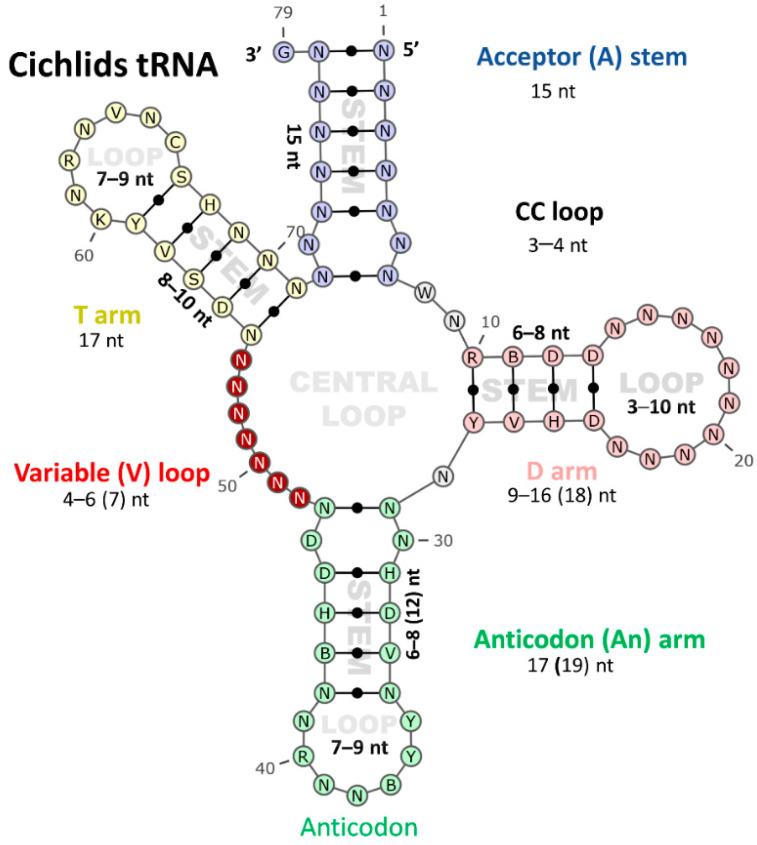
The tRNA secondary structures model represents the consensus sequence of 21 tRNAs in cichlid species. Each stem-loop is shown with a specific color. Arms of tRNAs (clockwise from top) are the amino acid acceptor stem (light purple), the dihydrouridine arm (light red), the anticodon arm (green), variable loop (dark red), and the thymidine arm (light yellow). The secondary structure model includes the standard tRNA numbering. The minimum and maximum length of each domain (stem and loop) are shown, and the numbers between brackets represent length with gaps. Degenerated nucleotides are used.

**Figure 3 biology-11-01522-f003:**
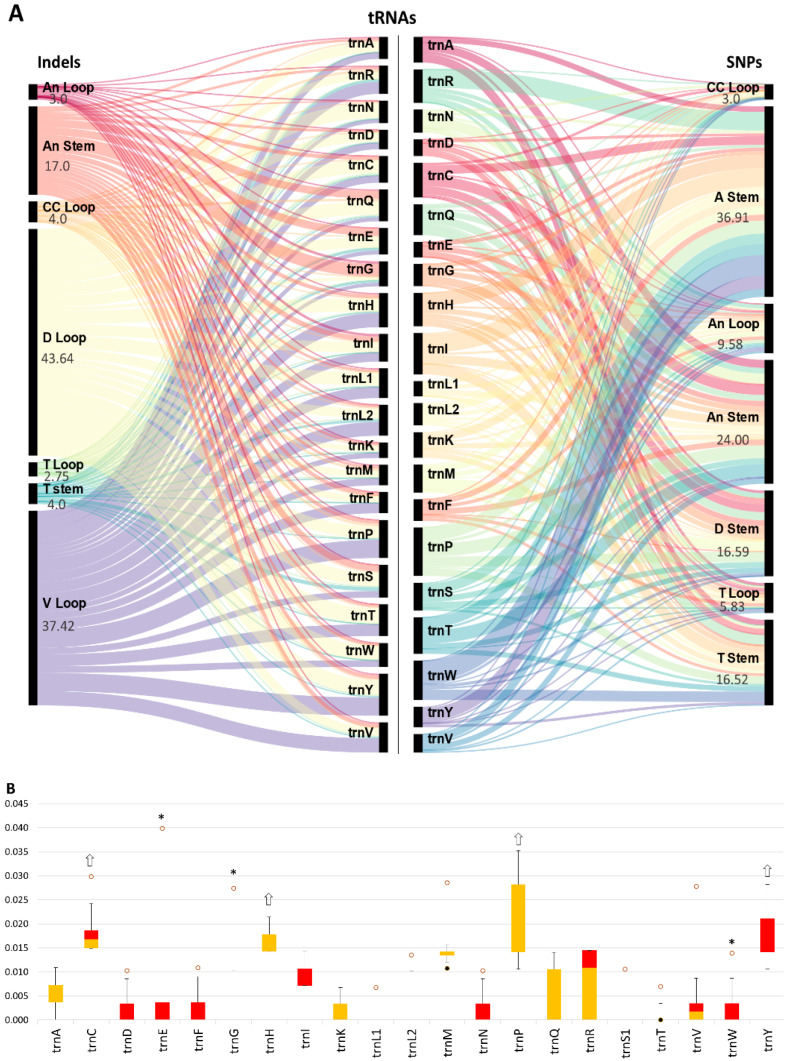
(**A**) An alluvial diagram showing the correlation between each tRNA and the structural distribution of tRNA mutations. Here, variables are assigned to vertical axes that are parallel, and the vertical columns represent 21 tRNAs (middle), the number of SNPs in each domain of tRNA (right), and the number of indels in each domain of tRNA (left). Mutations in each domain of tRNA structure cluster together, occupy a row in the diagram and are horizontally connected to the different tRNAs. Abbreviations of each domain: acceptor stem (A-stem), anticodon arm (An), dihydrouridine arm (D), thymidine arm (T), variable loop (V-loop), and the central connection loop (CC-loop). (**B**) The box plot displays the prevalence rate of the variation for each tRNA structure, for 21 mt tRNA between cichlids species. Boxes show median, 1st, and 3rd quartiles, and the whiskers represent the minimum and maximum values. Dots represent the minimum outlier (black), and the maximum outlier (orange). The significant Tajima D is indicated by (*). The white arrows indicate hypervariable tRNAs. The trnS1 is trnS^UGA^, trnL1 is trn^UAA^ and the trnL2 is trnL^UAG^.

**Figure 4 biology-11-01522-f004:**
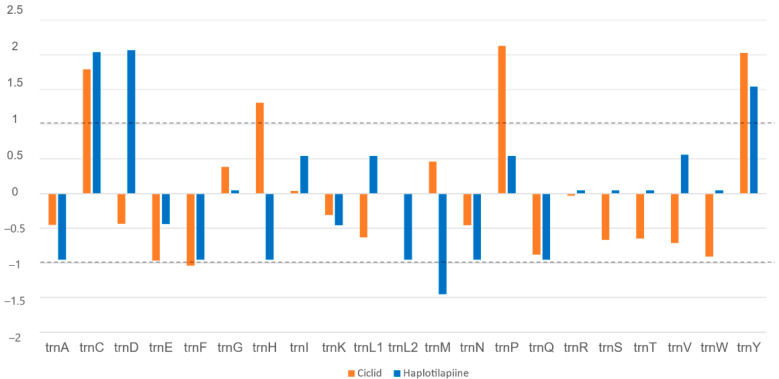
The total mutation rate in each tRNA gene in all cichlids compared to haplotilapiine lineage. Every two columns represent one of the tRNA genes. Orange color represents tRNAs in cichlid species, while blue represents the haplotilapiine species. Numbers above the z-score standard limits (dotted line; +1 or −1) represent the extreme hypervariable tRNA genes, while those below represent the extreme hypovariable tRNA genes. The trnS is trnS^UGA^, trnL1 is trn^UAA^ and the trnL2 is trnL^UAG^.

**Figure 5 biology-11-01522-f005:**
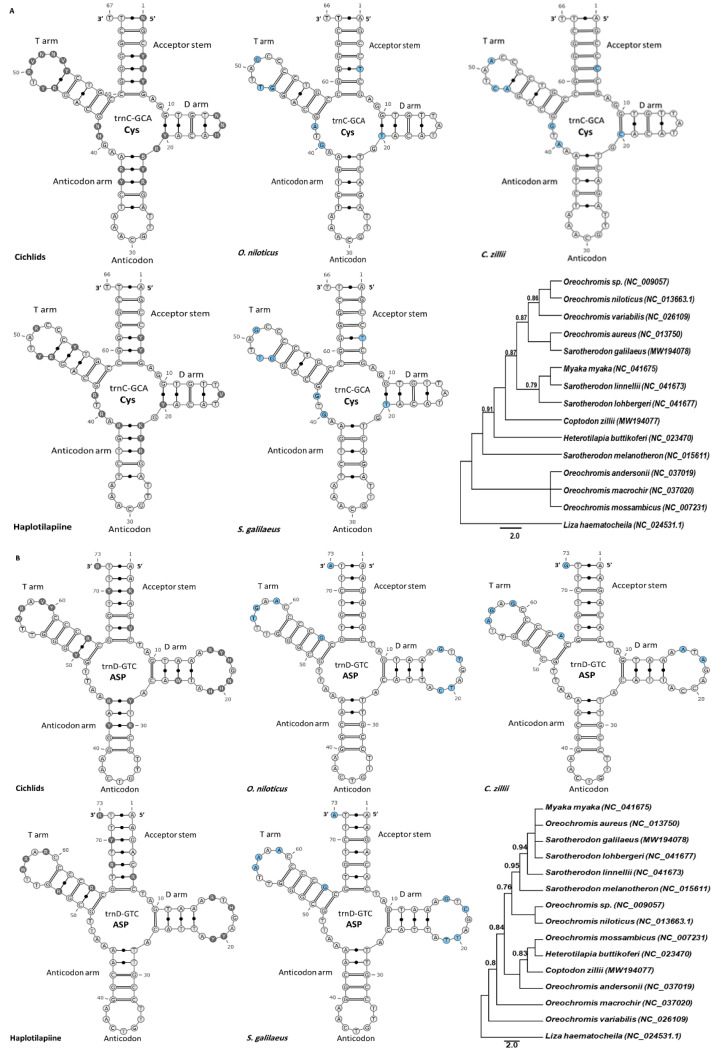
Superimposition of secondary structures of mitochondrial trnC (**A**) and trnD (**B**) molecules, showing the distribution of polymorphic sites among cichlids, haplotilapiine, and three species represents haplotilapiine species (*O. niloticus*, *C. zillii*, and *S. galilaeus*) abundant in the Nile River and connected waterbodies in Egypt. The tRNA genes are labeled with the abbreviations of their corresponding amino acids. The numbers indicate the position in the structure. Degenerated nucleotides are used, which is represented as a colored circle. The polymorphic nucleotides observed in cichlids and haplotilapiine are gray circles, while light blue circles correspond to polymorphisms in each species. Maximum likelihood phylogenetic tree based on trnC (**A**) and trnD (**B**) of haplotilapiine group. *Liza haematocheila* served as an outgroup species. Numbers beside internal branches indicate bootstrap value. GenBank accession numbers are listed between brackets.

**Figure 6 biology-11-01522-f006:**
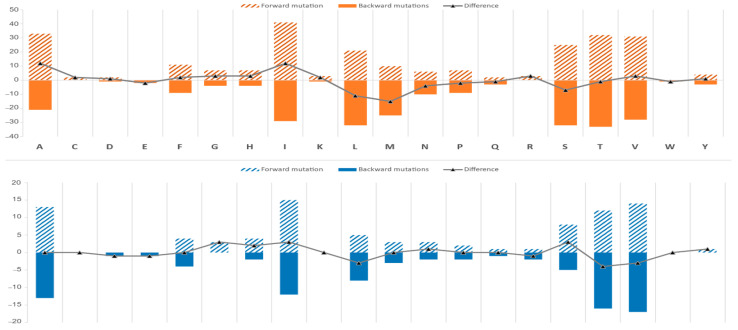
Observed amino acid changes in cichlid species (orange color) and haplotilapiine group (blue color). Forward mutations are labeled by pattern fill, leading to an increase in the amino acid. The backward mutations are labeled by the solid fill, leading to a decrease in the amino acid. The curved line represents the difference between forward and backward mutation. The amino acids are described with a one-letter abbreviation.

## Data Availability

DNA sequences were deposited in the NCBI GenBank database under accession number ON009397–ON009402.

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
