# Peer review of "The Evolutionary Dynamics of the Mitochondrial tRNA in the Cichlid Fish Family"

_biology, 2022, doi:10.3390/biology11101522_

Round 1

Reviewer 1 Report

The objective of the study presented by Fiteha and Magdy is to detect genetic variation of mitochondrial tRNAs
in the family Cichlidae and to study the impact of these variations
on the secondary structure of the aforementioned tRNAs. In addition, the authors aim to evaluate variable tRNAs and draw conclusions with respect to evolutionary prospects in the family Chichlidae. Although this work is quite clear and well-written, there are some parts that are missing and at least require minor revisions.

1) The authors talk about tRNAs having the sole function of transporting amino acids suitable for translation in eukaryotes. However, there are recent articles that show that eukaryotic tRNAs may also have other roles. This topic could be discussed further.
2) The difference between the role of tRNAs within mitochondria and the other nuclear tRNAs could also be better discussed.
3) An analysis of the genetic material of tRNAs at the nuclear DNA level would also be interesting, if the nuclear material is still available to the authors.
4) It's not clear to me what kind of filtering was used to discriminate between GenBank files to be used for analysis and those deleted. Number of cds? Number of rRNAs? Number of tRNAs? Were files including genome shotgun eliminated? 
5) It is not explained in detail which method/tool/algortim has been used to generate the secondary structure of the tRNAs that is the basis of this work since there are several types and they often give different results. Has a comparison been made between the various algorithms that generate RNA secondary structure?
6) The comparison with nuclear tRNAs is missing. It would be very interesting to have such an analysis at least using a reference genome.

7) Blast filters should be mentioned in the methods.
8) In total, the authors find different levels of RNA modifications at different positions especially  in  trnD, trnL, trnR, trnS, trnT, trnV, trnW, etc. tRNAs of haplotilapiine subfamily. They mention that their results provides an insight into the  tRNA evolution of the fish family analyzed and its effect on the cichlid diversity and speciation model. However, I think that in order to  provide insight into the molecular mechanisms underlying the decoding system and help to elucidate the molecular evolution of the whole family, it is necessary to compare their results to those of other authors that made similiar studies. Such a comparison in the discussion should be very important for this project.

Author Response

Dear Reviewer,

Thank you very much for your time and effort revising the manuscript, we have received your comments with great interest, please find our response to them as follows:

The objective of the study presented by Fiteha and Magdy is to detect genetic variation of mitochondrial tRNAs in the family Cichlidae and to study the impact of these variations on the secondary structure of the aforementioned tRNAs. In addition, the authors aim to evaluate variable tRNAs and draw conclusions with respect to evolutionary prospects in the family Chichlidae. Although this work is quite clear and well-written, there are some parts that are missing and at least require minor revisions.

1) The authors talk about tRNAs having the sole function of transporting amino acids suitable for translation in eukaryotes. However, there are recent articles that show that eukaryotic tRNAs may also have other roles. This topic could be discussed further.

Response: Thank you very much for the comment, we would like to confirm that the function of the tRNA to transport amino acid was specified during the gene translation process, please check line 67. However, the illustrated point was added to the introduction line 67.

2) The difference between the role of tRNAs within mitochondria and the other nuclear tRNAs could also be better discussed.

Response: We appreciate the suggestion very much. To add the nuclear tRNA is a great idea, however, we consider it as an additional scope to the mt tRNA evolution, especially when the complicity of nuclear-mitochondrial transfer DNA persists. Thus, it will require a different kind of analysis and sufficient data to be present for the whole cichlid family, which is not available as much as the mitochondrial genomes. We currently are performing whole-genome sequencing for the haplotilapiine species mentioned in the manuscript and will definitely consider the suggested point with proper analysis and adequate discussion. However, we had highlighted the possible difference at the structure level and also the association of DNA polymorphism to the pathogenesis and syndromes in humans (Lines 75:79).

3) An analysis of the genetic material of tRNAs at the nuclear DNA level would also be interesting, if the nuclear material is still available to the authors.

Response: As we responded above, the presence of enough nuclear genomes of cichlids to perform comprehensive analysis on the family is not as much as the available mitochondrial genomes.

4) It's not clear to me what kind of filtering was used to discriminate between GenBank files to be used for analysis and those deleted. Number of cds? Number of rRNAs? Number of tRNAs? Were files including genome shotgun eliminated?

Response: Please kindly note in lines 194 to 196 we explained the filtration features. I quote “17 were removed from the compilation due to significant variations regarding the other sequences, or annotation errors identifying tRNA genes and/or being identified as a recombinant sequence”. The deviated sequences and recombinants were directly discarded, while the mitogenomes with wrong annotated tRNA were reannotated, but some mitogenomes failed to be reannotated at tRNA level and thus were discarded.

5) It is not explained in detail which method/tool/algorithm has been used to generate the secondary structure of the tRNAs that is the basis of this work since there are several types and they often give different results. Has a comparison been made between the various algorithms that generate RNA secondary structure?

Response: Please kindly note that a separate paragraph between lines 104 to 110 describes the tRNA annotation including all the details for the running parameters. We used tRNAscan-SE, a highly cited software (cited 1505 times for the 1st version and 445 times for the 2nd version since 2016 and 2019, respectively) and that is included as one of the frequently used tools for organelles tRNA annotation. The tool provides both the annotation and the secondary structure based on the published data; thus, the outputs are more by similarity than by prediction, which provides a more solid base for tRNA secondary structure formation than those generated by prediction. Regarding the comparison with other software, we initially included other tRNA annotators in our re-annotation process (e.g., ARAGORN v1.2.38 and ARWEN v1.2.3) and RNAfold for tRNA secondary structure folding, however, the only software that combined both annotation and secondary structure formation was tRNAscan-SE, a more tRNA oriented software than RNAfold.

6) The comparison with nuclear tRNAs is missing. It would be very interesting to have such an analysis at least using a reference genome.

Response: Thanks again for the suggestion. As previously responded, not enough complete genomic assemblies for the cichlid family is available. Additionally, a reference genome should provide a specific variation of the reference species which will not be adequate to compare with consensus tRNAs formed by the whole cichlid family, as the level of variation will be incompatible. However, according to our plan, we will consider the suggestion at least to compare the mt-tRNAs with the nuclear tRNAs of the haplotilapiine species when the genome assembly is finished and ready for reporting.

7) Blast filters should be mentioned in the methods.

Response: Thanks for the comment, we didn’t apply any filters and used the default BLAST search parameters. Thus, we added to the M&M part line 187: “applying default parameters”.

8) In total, the authors find different levels of RNA modifications at different positions especially in trnD, trnL, trnR, trnS, trnT, trnV, trnW, etc. tRNAs of haplotilapiine subfamily. They mention that their results provide an insight into the tRNA evolution of the fish family analyzed and its effect on the cichlid diversity and speciation model. However, I think that in order to provide insight into the molecular mechanisms underlying the decoding system and help to elucidate the molecular evolution of the whole family, it is necessary to compare their results to those of other authors that made similar studies. Such a comparison in the discussion should be very important for this project.

Response: Thank you very much for the feedback on the study objective. We agreed that the statement was more generalist than it should be. The insights would be specifically for the mitochondrial tRNA evolution and its effect on the cichlid diversity and speciation model at the maternal level. Thus, we will change the statement in the abstract and in the discussion parts to be more specific. Regarding the comparison with similar studies, kindly note that we included several references concerned with the mt-tRNA analysis, some were on human, metazoan (in general), acariforms, and scorpions (lines no. 444 to 452); however, after your suggestion we improved the sentence to clarify the plants and fungi difference at the D-armless trnS gene (line 446).

Reviewer 2 Report

The authors describe the evolutionary variations of mitochondrial tRNAs of the cichlid fish family. These fishes are prevalent in Africa and famous for their rapid speciation and high-level phenotypic diversity. Contrary to one of the authors' hopes, written as "The current study provides ...the effect (of tRNA evolution) on the cichlid diversity and speciation model (l. 44)", nothing was found so unusual in the mit-tRNA variation that the hyper-diversity of cichlids could be explained. However, the presented data is valuable on their own, and I believe that the manuscript is worth publishing after they addressed my concerns below.

(l. 16) "a single gene was defected"

I suppose that this defective gene is trnS(GCU) lacking D domain. "Defect" implies "non-functional". I wonder if the authors have any evidence that this tRNA is not functional, except for its missing D arm. The authors should not have omitted analyses of this trnS(GCU) in their study. Even if this tRNA was not functional, its analysis would be interesting because its mutation rate can be compared with those of other functional tRNAs.

(L. 21) "Tryptophan tRNA carrier"

This is a confusion between "Tryptophan tRNA" and "Tryptophan carrier (=Trp tRNA); it should be "Tryptophan tRNA".

Be specific about "signatures of possible selection" (L. 21).

The authors contrast trnP and trnL(UAA) in the sentence, "Most of the mutations were ...(L. 32)".  I wonder what kind of contrast they make with each other. Since trnP is the most highly variable in its sequence, the contrasted trnL(UAA) should be the least variable. However, the least variable molecule is trnW as the authors write (L. 35). 

The terms "neutral" and "unneutral" should be explained (L. 35, 39, and others). It seems that “neutral” tRNAs can accumulate neutral mutations in them and vary in the base sequence, as opposed to “unneutral” tRNAs being highly conserved in their sequence. This understanding is right?

I do not understand the structures of two sentences: "From a structural perspective, formations... (L. 32-35)" and "The trnW was the lowest....in contrast to the neutral trmH... (L. 35-38)".

I do not understand either the sentence, "In CDS, the highest affected amino acid... (L. 42-43).

With these problems, the contents of page 1 (Simple Summary and Abstract) can hardly be followed without referring to the succeeding pages. I think that these two parts should be rewritten to better convey the findings of this study.

Minor points.

"A-stem" is a shortened form of "Acceptor stem, and "An" means "anticodon", as described in L. 116. These shortened terms should be indicated in Fig. 2 beside the secondary structure of tRNA for the convenience of the reader.

The authors write "trnL1" and "trnL2" in Figs. 3 and 4. They are trnL(TAA or UAA) and trnL(UAG) in the text and should be named the same way in the text and figures.

Explain "Tajima D values" to allow the reader to understand Fig.4.

Author Response

Dear Reviewer,

Thank you very much for your time and effort revising the manuscript, we have received your comments with great interest, please find our response to the raised concerns as follows:

The authors describe the evolutionary variations of mitochondrial tRNAs of the cichlid fish family. These fishes are prevalent in Africa and famous for their rapid speciation and high-level phenotypic diversity. Contrary to one of the authors' hopes, written as "The current study provides ...the effect (of tRNA evolution) on the cichlid diversity and speciation model (l. 44)", nothing was found so unusual in the mit-tRNA variation that the hyper-diversity of cichlids could be explained. However, the presented data is valuable on their own, and I believe that the manuscript is worth publishing after they addressed my concerns below.

(l. 16) "a single gene was defected"

I suppose that this defective gene is trnS(GCU) lacking D domain. "Defect" implies "non-functional". I wonder if the authors have any evidence that this tRNA is not functional, except for its missing D arm. The authors should not have omitted analyses of this trnS(GCU) in their study. Even if this tRNA was not functional, its analysis would be interesting because its mutation rate can be compared with those of other functional tRNAs.

Response: Thank you very much for the comment, we are sorry to misuse the word defect, we meant deformed, as the armless or bizarre tRNAs can be functional. Therefore, we changed the word “defect” to “D-armless”. Regarding the inclusion of the trnS (GUS), we have initiated our analysis including it to the whole tRNA sets, however, the variation in D-arm was extremely variable compared to other formations. When we excluded it, we had the chance to measure the polymorphism in the D-arm with better resolution and more informative way. Thus, we decided to exclude and report its deformity or D-armless status separately without affecting the other tRNAs.  

(L. 21) "Tryptophan tRNA carrier"

This is a confusion between "Tryptophan tRNA" and "Tryptophan carrier (=Trp tRNA); it should be "Tryptophan tRNA".

Response: Noted and changed accordingly.

Be specific about "signatures of possible selection" (L. 21).

Response: Thank you for pointing this out, we specified the selection to be: signatures of possible purifying selection.

The authors contrast trnP and trnL(UAA) in the sentence, "Most of the mutations were ...(L. 32)".  I wonder what kind of contrast they make with each other. Since trnP is the most highly variable in its sequence, the contrasted trnL(UAA) should be the least variable. However, the least variable molecule is trnW as the authors write (L. 35).

Response: Thank you very much for the observation. The phrase required clarification as pointed. In the comparison between the different consensus tRNAs, the trnP and trnL (UAA) were the highest and lowest polymorphic consensus tRNAs, respectively. However, the trnW was the least variable among cichlids and among haplotilapiine species when polymorphism was measured for each tRNA independently.

The terms "neutral" and "unneutral" should be explained (L. 35, 39, and others). It seems that “neutral” tRNAs can accumulate neutral mutations in them and vary in the base sequence, as opposed to “unneutral” tRNAs being highly conserved in their sequence. This understanding is right?

Response: In a simple sense, yes. The neutral genes are affected by random mutations, while unneutral are affected by non-random ones. We believe that the explanation was already given in lines 472 to 484. We are willing to change or add any additional suggestions to simplify the term in the next round if the stated explanation is not clear enough.

I do not understand the structures of two sentences: "From a structural perspective, formations... (L. 32-35)" and "The trnW was the lowest....in contrast to the neutral trmH... (L. 35-38)".

Response: We are very sorry, the sentence in L32-35 is missing additional information, it is a writing mistake, that mostly occurred during the track cleaning after our revisions. The sentence should be: From a structural perspective, the anticodon loop and T-loop formations were the most conserved structures among all parts of the tRNA in contrast to the A-stem and D-loop formations.

We improved the sentence in L35-38 to become: The trnW was the lowest polymorphic unneutral tRNA among all cichlids (both the family and the haplotilapiine lineage), in contrast, the neutral trnD that was extremely polymorphic among and within the haplotilapiine lineage species compared to other cichlids species.

I do not understand either the sentence, "In CDS, the highest affected amino acid... (L. 42-43).

Response: We rephrased the sentence to clarify the meaning and the purpose of the sentence, to be: By observing the DNA polymorphism in the coding DNA sequences (CDS), the highest affected amino acid by non-synonymous mutations was Isoleucine and was equally mutated to Valine and vice versa; no correlation between mutations in CDS and tRNAs was statistically found.

With these problems, the contents of page 1 (Simple Summary and Abstract) can hardly be followed without referring to the succeeding pages. I think that these two parts should be rewritten to better convey the findings of this study.

Response: Thank you very much for the suggestion. After following your previous suggestion, we think those parts are more independent than the text now. However, we are open to more suggestions to improve the independence and content of those two sections.

Minor points:

"A-stem" is a shortened form of "Acceptor stem, and "An" means "anticodon", as described in L. 116. These shortened terms should be indicated in Fig. 2 beside the secondary structure of tRNA for the convenience of the reader.

Response: Thank you for the comment, the point was added as suggested.

The authors write "trnL1" and "trnL2" in Figs. 3 and 4. They are trnL(TAA or UAA) and trnL(UAG) in the text and should be named the same way in the text and figures.

Response: Thank you very much for the comment, we worked out an alternative solution to note the trnL number with the trnL type in the figure legend, to reduce the working time on the adjustment of the figures.

Explain "Tajima D values" to allow the reader to understand Fig.4.

Response: Thank you very much for the suggestion, we believe the understanding of Fig. 4 is not dependent on the Tajima D explanation, the tRNA with significant D values were marked with (*). We added a statement to the results text (line 329): The Tajima D values ranged from 0.0383 to -2.069 and were found to be "significantly deviated from zero"…etc.

The deviation from zero indicates the presence of non-random factors. The reason is explained in the discussion part lines 472:484.